

# Growth and feeding ecology of coniform conodonts

Isabella Leonhard[1], Bryan Shirley[2], Duncan J. E. Murdock[3],
John Repetski[4] and Emilia Jarochowska[5]

[1] Institute of Evolutionary Biology, University of Warsaw, Warsaw, Poland
[2] Paläoumwelt, Friedrich-Alexander Universität Erlangen-Nürnberg, Erlangen, Bavaria, Germany
[3] Oxford University Museum of Natural History, Oxford, United Kingdom
[4] US Geological Survey-Emeritus, Reston, Virginia, United States of America
[5] Department of Earth Sciences, Utrecht University, Utrecht, Netherlands

Corresponding author
Isabella Leonhard,
isabella.leonhard@fau.de

## ABSTRACT

Conodonts were the first vertebrates to develop mineralized dental tools, known as elements. Recent research suggests that conodonts were macrophagous predators and/or scavengers but we do not know how this feeding habit emerged in the earliest coniform conodonts, since most studies focus on the derived, 'complex' conodonts. Previous modelling of element position and mechanical properties indicate they were capable of food processing. A direct test would be provided through evidence of *in vivo* element crown tissue damage or through *in vivo* incorporated chemical proxies for a shift in their trophic position during ontogeny. Here we focus on coniform elements from two conodont taxa, the phylogenetically primitive *Proconodontus muelleri* Miller, 1969 from the late Cambrian and the more derived *Panderodus equicostatus* Rhodes, 1954 from the Silurian. Proposing that this extremely small sample is, however, representative for these taxa, we aim to describe in detail the growth of an element from each of these taxa in order to the test the following hypotheses: (1) *Panderodus* and *Proconodontus* processed hard food, which led to damage of their elements consistent with prey capture function; and (2) both genera shifted towards higher trophic levels during ontogeny. We employed backscatter electron (BSE) imaging, energy-dispersive X-ray spectroscopy (EDX) and synchrotron radiation X-ray tomographic microscopy (SRXTM) to identify growth increments, wear and damage surfaces, and the Sr/Ca ratio in bioapatite as a proxy for the trophic position. Using these data, we can identify whether they exhibit determinate or indeterminate growth and whether both species followed linear or allometric growth dynamics. Growth increments (27 in *Pa. equicostatus* and 58 in *Pr. muelleri*) were formed in bundles of 4–7 increments in *Pa. equicostatus* and 7–9 in *Pr. muelleri*. We interpret the bundles as analogous to Retzius periodicity in vertebrate teeth. Based on applied optimal resource allocation models, internal periodicity might explain indeterminate growth in both species. They also allow us to interpret the almost linear growth of both individuals as an indicator that there was no size-dependent increase in mortality in the ecosystems where they lived *e.g.*, as would be the case in the presence of larger predators. Our findings show that periodic growth was present in early conodonts and preceded tissue repair in response to wear and damage. We found no microwear and the Sr/Ca ratio, and therefore the trophic position, did not change substantially during the lifetimes of either individual. Trophic ecology of coniform conodonts differed from the predatory

and/or scavenger lifestyle documented for "complex" conodonts. We propose that conodonts adapted their life histories to top-down controlled ecosystems during the Nekton Revolution.

# INTRODUCTION

Conodonts are marine, eel-like jawless vertebrates occurring in marine ecosystems from the Cambrian to the Late Triassic. As the first marine vertebrates possessing a mineralized skeleton (*Sweet & Donoghue, 2001*), conodonts receive special attention from researchers for several reasons. Their oropharyngeal cavity contained an array of phosphatic dental tools known as elements. Biostratigraphic and geochemical studies are commonly conducted on these elements, thanks in part to their abundance in marine carbonates, complex morphology, and geochemical stability (*Boaz, Kolodny & Kovach, 1984*; *Trotter et al., 2007*; *Joachimski & Buggisch, 2002*; *Joachimski et al., 2009*; *Katvala & Henderson, 2012*). In spite of their utility, the ecology and feeding strategies of conodonts remain enigmatic. Elements range from simple cone-shaped, or 'coniform', morphologies in more primitive groups to comb-like and platform-bearing ('complex') forms, which are arranged in bilaterally symmetrical multi element feeding apparatus (*Aldridge et al., 1993*). Their highly diverse morphology strongly indicates an enormous range of feeding ecologies (*Girard & Renard, 2012*; *Murdock, Rayfield & Donoghue, 2014*; *Martínez-Pérez et al., 2016*; *Petryshen et al., 2020*). Furthermore, conodonts provide a great resource to identify general functional principles of the evolution of dental tools at the biomechanical, morphological and histological level (*Jones, 2009*; *Jones et al., 2012a*, *2012b*; *Martínez-Pérez et al., 2014*; *Dzik, 2015*; *Martínez-Pérez et al., 2016*; *Guenser et al., 2019*; *Petryshen et al., 2020*). Multiple modes of feeding have been proposed in the past, but only the macrophagous predator or scavenger (*Aldridge et al., 1986*; *Purnell, 1993*) and filtering as a microphagous active suspension feeder (*Nicoll & Rexroad, 1987*) are compatible with the conodont body plan (*Purnell & Donoghue, 1997*; *Donoghue, Purnell & Aldridge, 1998*). These interpretations have been established in quantitative studies on growth dynamics (*Armstrong & Smith, 2001*; *Zhan, Aldridge & Donoghue, 1997*), histology (Donoghue, 1997) and chemical composition (*Shirley et al., 2018*; *Balter et al., 2019*) carried out mostly on "complex" conodonts. However, there is little evidence how feeding strategies evolved within the very early coniform conodonts (*Murdock, Sansom & Donoghue, 2013*; *Murdock, Rayfield & Donoghue, 2014*). Here we attempt to pinpoint the assembly of morphogenetic and life history adaptations at the origin of predation within the earliest vertebrates.

## Apparatus reconstruction

The coniform conodont genus *Panderodus* is regarded by most researchers to be the only known coniform taxon represented by almost complete fused clusters (*An et al., 1983*; *Kozur, 1984*; *Dzik & Drygant, 1986*) and natural assemblages (*Smith, Briggs & Aldridge,*

*1987*; *Murdock & Smith, 2021a*). Multiple reconstructions of its apparatus have been proposed (*e.g. Fåhræus & Hunter, 1985*; *Dzik & Drygant, 1986*; *Armstrong, Clarkson & Owen, 1990*; *Sansom, Armstrong & Smith, 1994*), with the most recent synthesis proposed by *Murdock & Smith (2021a)*. According to the reconstruction of *Sansom, Armstrong & Smith (1994)* and *Murdock & Smith (2021a)*, the apparatus of *Panderodus* consists of 17 cone-shaped, laterally furrowed and non-geniculate elements assigned to six morphotypes arranged symmetrical across the midline. *Sansom, Armstrong & Smith (1994)* subdivided the elements into three architectural units. The costate suite anterior of the apparatus comprises arcuatiform, graciliform and truncatiform elements. The compressed posterior suite consists of falciform and tortiform elements and the last unit is the unpaired symmetrical aequaliform element on the midline. The most recent reconstruction of *Murdock & Smith (2021a)* allows to identify homologies with feeding apparatuses of more derived conodonts, with differentiation between grasping M and S elements (rostrally) and caudally located P elements for food processing (*Aldridge et al., 1993*). According to the newest data, the costate element suite consists of arcuatiform, as well as of four pairs of graciliform elements. The compressed, caudal suite comprises falciform, tortiform and truncatiform elements, while the unpaired aequaliform element exposes at the midline between costate and compressed suite and not at the very caudal end of the apparatus as proposed earlier.

Apparatus reconstructions of the genus *Proconodontus* are more hypothetical as they are not based on natural assemblages and/or clusters. *Proconodontus muelleri* Miller, 1969 exhibits a trimembrate apparatus, comprising symmetrical (aequaliform), asymmetrical (graciliform) and compressed (arcuatiform) morphotypes (*Szaniawski & Bengston, 1998*).

## Function of coniform conodont elements

*Jeppsson (1979)* discussed the use of "simple type conodonts" as teeth based on morphological similarities between actual teeth and conodont elements. *Sansom, Armstrong & Smith (1994)* proposed functional differentiation within the apparatus of coniform conodonts because of morphological differences of single element morphotypes, and *Szaniawski (2009)* proposed that some coniform conodonts, including *Panderodus*, were venomous. This classification was corroborated by *Murdock, Sansom & Donoghue (2013)* through the development of functional interpretations for each morphotype, using biomechanical proxies to infer their relative propensity for a cutting or grasping function.

## Internal structure

Euconodont (or "true conodonts") elements consist of hypermineralized crown tissue and the dentine-like basal body (*Bengtson, 1976*). The basal body is organic-rich, poorly mineralized and rarely preserved (*Lindström, 1965*; *Souquet & Goudemand, 2020*). The crown tissue is composed of hyaline lamellar tissue and, in some conodont taxa, white matter, which is unique to conodonts (*Pander, 1856*; *Hass, 1941*; *Müller & Nogami, 1971*). The lamellar tissue consists of individual growth layers that accrete appositionally throughout the life of the animal (*Müller & Nogami, 1971*; *Donoghue, 1998*).

The enamel-like structure of the lamellar tissue has been proposed to be an adaptation to dental function (*Donoghue, 2001*), a hypothesis supported by Finite Element Analyses of early coniform conodonts (*Murdock, Rayfield & Donoghue, 2014*). It has been suggested that white matter is a further adaptation to dental function by allowing the elements to withstand greater tensile stresses (*Jones et al., 2012a*).

## Element growth

Conodont elements grew by periodic, appositional accretion of new lamellar crown increments (*Bengtson, 1976*; *Donoghue, 1998*), speculated to reflect daily periods of growth (*Zhan, Aldridge & Donoghue, 1997*; *Dzik, 2008*; *Świś, 2018*). Deposition of lamellae in bundles, *i.e.* periodicity, within conodont crown tissue has been observed in "complex" conodonts (*Zhan, Aldridge & Donoghue, 1997*; *Chen et al., 2016*), as well as in coniforms (*Armstrong & Smith, 2001*). *Shirley et al. (2018)* showed that these bundles correspond to periods of repair after surface damage resulting from food processing. The distribution of such abraded and truncated surfaces on elements corresponds to patterns of microwear distribution observed on their surfaces (*Purnell, 1995*; *Jones et al., 2012b*). Alternatively, such internal discontinuities have been interpreted as structures resulting from accidental damage followed by repair (*Hass, 1941*) or abnormal deformation during growth (*Rhodes & Phillips, 1954*). *Purnell (1995)* interpreted them as evidence for phases of growth and function based on consistent occurrence, which was corroborated by *Donoghue (1998)*. Distinct growth dynamics and morphology between the early ontogenetic and adult phase have been observed in several conodont taxa (*Armstrong & Smith, 2001*) and further supported by differences in chemical composition (*Shirley et al., 2018*).

## Chemical proxies for the trophic position

Conodont chemical composition has been determined by *Pietzner et al. (1968)* as consistent with francolite and close to the non-stoichiometric formula $Ca_5Na_{0.14}(CO_3)_{0.16}(PO_4)_{3.01}(H_2O)_{0.85}F_{0.73}$ (*Joachimski et al., 2009*). In addition, alkali-earth elements (Sr, Ba and Mg), as well as divalent metals, substitute Ca, Na and F *in vivo* (*Reynard & Balter, 2014*). Rare Earth elements (REE) and high-field strength elements are incorporated biostratinomically (*Wright, Seymour & Shaw, 1984*; *Trotter & Eggins, 2006*; *Reynard & Balter, 2014*; *Žigaitė et al., 2020*). As a nonessential element, strontium is increasingly depleted relative to the essential element Ca at each transition to a higher level in the trophic chain. This process is referred to as biopurification, resulting in lower Sr/Ca ratios with increasing trophic level (*Comar, Russell & Wasserman, 1957*; *Elias, Hirao & Patterson, 1982*). Strontium/calcium (Sr/Ca) ratio analysis of bone and teeth has been applied previously to investigate palaeodiets and relative positions of animals within the trophic network (*Balter et al., 2002*; *Peek & Clementz, 2012*). This proxy has been mostly applied in terrestrial environments (*e.g.*, *Sillen, 1992*; *Sillen & Lee-Thorp, 1994*; *Balter, 2004*; *Sponheimer et al., 2005*), whereas similar investigations in marine food webs are lacking. It has been used successfully in modern environments and, even though the impact of bioapatite diagenesis (*e.g. Ferretti et al., 2021*) on the preservation of

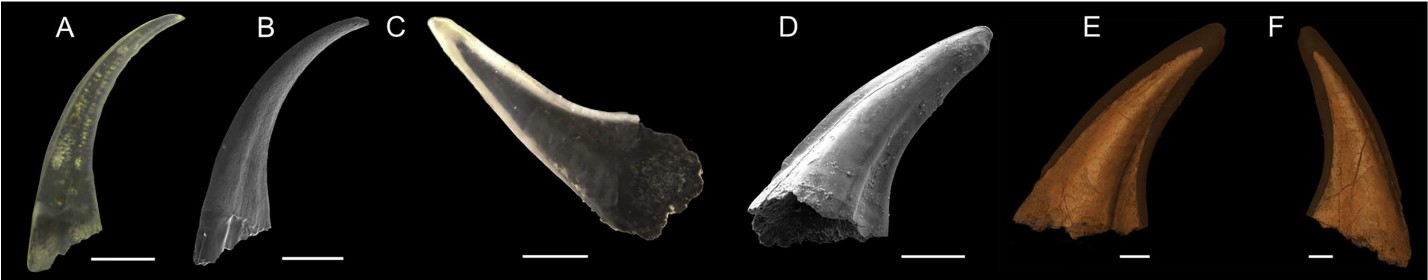

**Figure 1 Overview about the specimens used (Light microscope and SEM image, as well as SRXTM scan).** (A and B) Light microscope image (A) and SEM-image (B) of *Pa. equicostatus*; (C–F) Light microscope image (C), SEM-image (D) and SRXTM scans (E and F) of *Pr. muelleri*. Scale bars equal 200 μm (A–D) and 100 μm (E and F).

this proxy has not been investigated so far, it yields consistent results in fossil hypermineralized tissues (*e.g. Balter et al., 2002*; *Sponheimer et al., 2005*).

*Wright (1990)* demonstrated variations in Sr content within the lamellar tissue: light bands have higher Sr content than dark bands, which underpins visibility of the lamellae in BSE imaging. These observations were confirmed by several studies (*e.g. Zhuravlev & Shevchuck, 2017*; *Shirley et al., 2018*). Additionally, individual growth stages of the "complex" conodont *Ozarkodina confluens* are distinguished by a decrease in crown tissue Sr content (*Shirley et al., 2018*), coincident with the appearance of histological record of dental wear in the adult animal.

## Coniform conodonts

In this study, we focus on two coniform conodont species: the early *Proconodontus muelleri* Miller, 1969 (Proconodontidae: late Cambrian to Ordovician) and the more derived species *Panderodus equicostatus* Ethington, 1959 (Panderodontidae), which was widespread in Ordovician to Devonian oceans. Both species represent increased specialization of conodont apparatus during conodont evolution. Using backscatter electron (BSE) imaging and energy-dispersive X-ray spectroscopy (EDX), we test (1) if both conodont species processed hard food by analyzing whether their feeding behavior is manifest in damaged dental tissues; and (2) if both species shifted their trophic niche towards higher trophic levels during ontogeny, undergoing chemical and morphological changes in their crown tissues. Furthermore, we set out to identify whether these species show determinate or indeterminate growth and whether it follows linear or allometric dynamics, which might inform us on the evolution of life histories of these organisms. Growth periodicity has been previously observed in conodonts (*Armstrong & Smith, 2001*) and attributed to episodes of repair following intensive feeding periods by *Shirley et al. (2018)*. Here we aim to identify whether growth periodicity correlates with tissue damage and repair.

## MATERIALS AND METHODS

We used two coniform conodont elements (Fig. 1); one truncatiform element (Figs. 1A and 1B) of *Panderodus equicostatus Rhodes & Phillips, 1954* (for anatomical notation see *Sansom, Armstrong & Smith, 1994*) from the Homerian (middle Silurian) shallow marine

carbonates of the Ternava Formation at Vrublivtsy, Ukraine (sample V-19.25 in *Jarochowska et al., 2016*). It is stored in the collections of GeoZentrum Nordbayern (accession number EJ-12-V-19.25-001). The second specimen (Figs. 1C–1F) is an aequaliform element (for anatomical notation see *Müller, 1973*; *Miller 1980*) of *Proconodontus muelleri* Miller 1969 collected from Windfall Formation, Eureka County, Nevada, USA, dated at the *Eoconodontus* Zone, Furongian (Cambrian) (from sample 5-22-08D, collected by J.D. Loch and J.F. Taylor; see *Loch, Taylor & Repetski (2019)*; will be stored at the U.S. Geological Survey under the accession number DM_WI_Prc07).

## Synchrotron Radiation X-ray Tomographic Microscopy

Specimen [DM_WI_Prc07] (*Pr. muelleri*) was scanned at the TOMCAT X02DA beamline at the Swiss Light Source, Paul Scherrer Institute, Villigen, Switzerland. The sample was mounted on a 3 mm brass stub using an acetone soluble glue. A 20× objective, 17 keV energy, and exposure time of 300 ms were used for the scan acquiring 1501 individual projections. These were then reconstructed using a 60-core Linux PC farm which applied a Fourier transform routine and a regridding procedure as outlined by *Zhu et al. (2010)*. The subsequent model had voxel dimensions of 0.325 μm. Using Amira 2019, Slice data were segmented and cleaned to produce 3D models (Figs. 1E and 1F).

## Sample preparation

Both samples were prepared following the method outlined by *Shirley, Bestmann & Jarochowska (2020)*. Conodonts were imbedded in epoxy resin (Epofix, 398; Struers, Copenhagen, Denmark) and the surfaces were ground and polished to create a flat and defect-free surface. This was followed by carbon coating up to 7 nm in thickness. Specimen photographs are stored on Morphobank (morphobank.org: http://morphobank.org/permalink/?P3589link; Specimen no. M681817 and M821733).

## Energy-dispersive X-ray spectroscopy

Position of EDX transects with respect to conodont tissue was documented in BSE images obtained using a TESCAN Vega\\XMU scanning electron microscope at GeoZentrum Nordbayern, Friedrich-Alexander-Universität Erlangen-Nürnberg. EDX analysis was performed using an Oxford Instruments X-MAX 50 mm silicon drift detector. The concentrations of major constituents (Sr, F, Mg, P, Na, Ca, O) of the conodont elements were measured along three line transects (*Pr. muelleri*; Fig. 2, Table S1) and six line transects (*Pa. equicostatus*; Fig. 3, Table S2) using a voltage of 15 KeV with a spatial resolution of ~3 μm (*Shirley, Bestmann & Jarochowska, 2020*). EDX was calibrated using a cobalt standard. All line transects were run for at least 45 min, to quantify spatial trends of Sr and Ca concentration throughout basal body and crown tissue, as well as changes in the content of these elements in the crown lamellae. The elemental composition of single elements was measured as total number of counts (cts), with each point measured for the same time. This allows comparing relative Sr and Ca content between all spots in the line transect. We excluded measurements which fell into cracks within the element

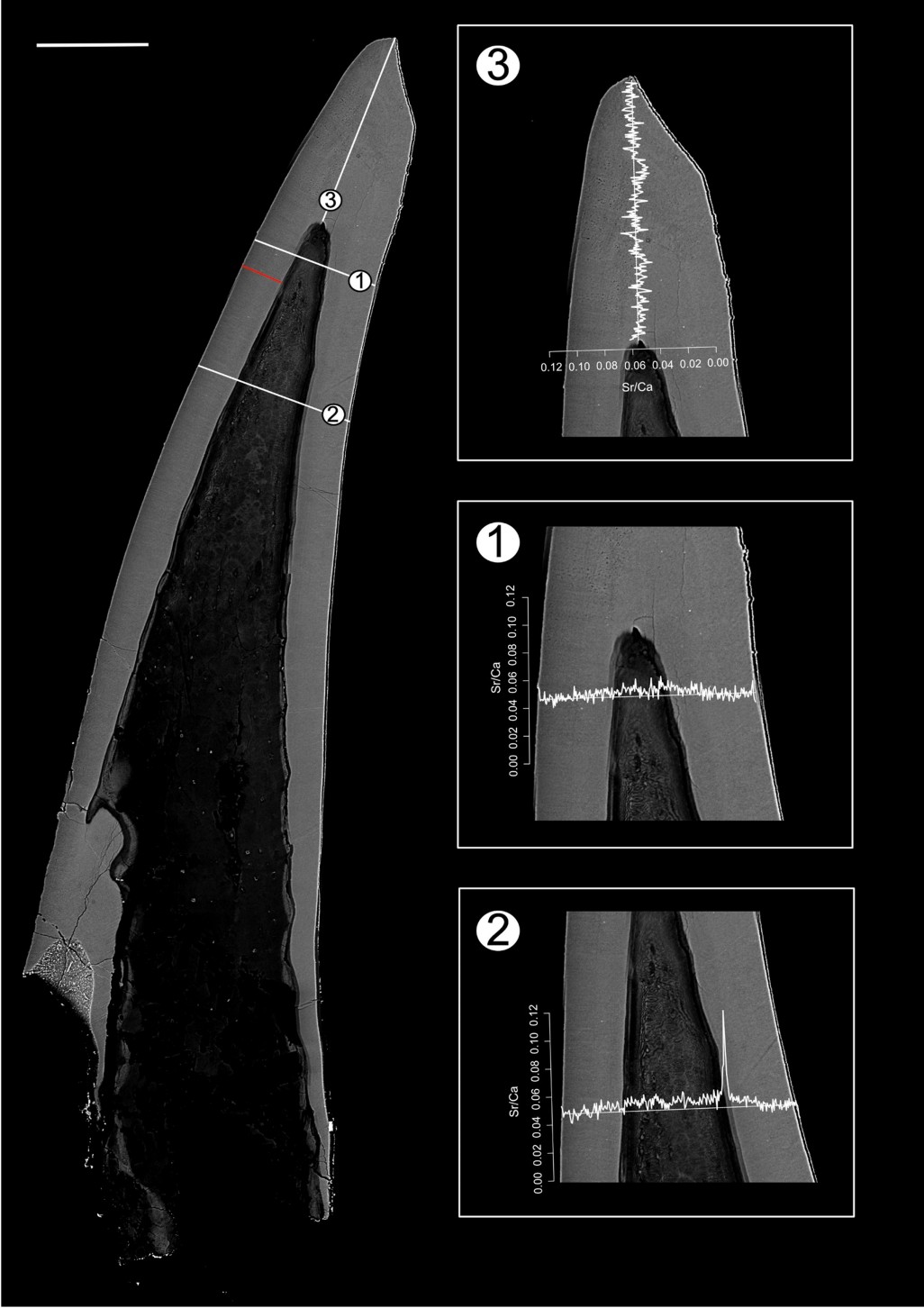

**Figure 2 BSE image of *Proconodontus muelleri* outlining the transects along which Sr and Ca contents were measured.** Composite BSE image of the polished aequaliform element of *Pr. muelleri* outlining three transects (1–3) along which Sr and Ca contents were measured. Changes in the Sr/Ca ratio with growth are expressed as the number of counts (cts). The red line marks the transect along which the lamellae were counted (for close-up see Fig. 6A). Scale bar equals 200 µm.

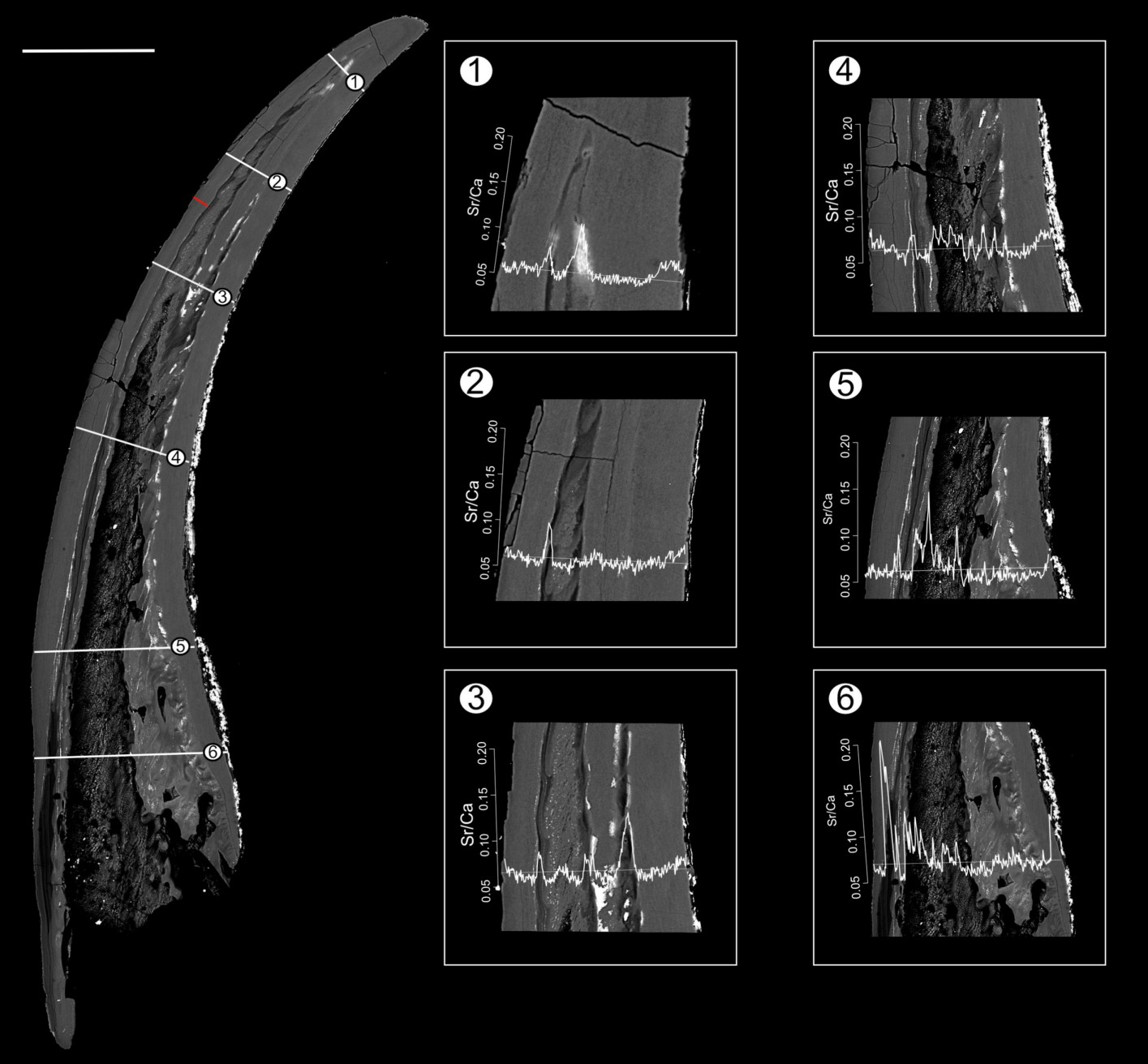

**Figure 3 BSE image of *Panderodus equicostatus* outlining the transects along which Sr and Ca contents were measured.** Composite BSE image of the polished truncatiform element of *Pa. equicostatus* outlining six transects (1–6) along which Sr and Ca contents were measured. Changes in the Sr/Ca ratio though ontogeny are expressed as the number of counts (cts). Transect six was excluded from the analysis since most of it lied within the basal body). The red line marks the transect along which the lamellae were counted (for close-up see Fig. 7A). Scale bar equals 200 μm.

or into the resin. Since basal tissue with its high organic content does not preserve chemical concentration reliably, we focused on the Sr and Ca content within the crown tissue (Fig. 4).

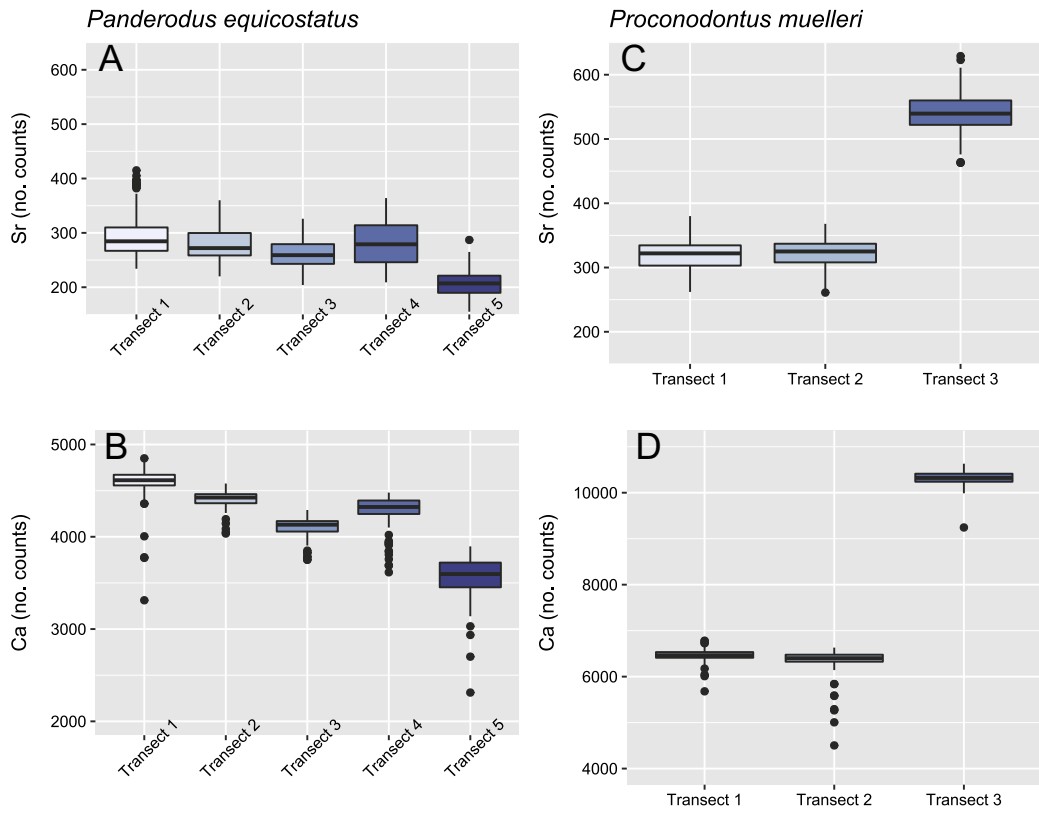

**Figure 4 Relative concentrations of Sr and Ca per transects in the crown tissue of *Panderodus equicostatus* and *Proconodontus muelleri*.** Relative concentrations of Sr and Ca per transect (247 points per transect) in the crown tissue of *Pa. equicostatus* (A and B) and *Pr. muelleri* (C and D).

All analyses were carried out using the R Software (*R Core Team, 2020*). A random slope and random intercept mixed-effects model was fitted in the lme4 package (*Bates et al., 2015*) to Sr/Ca values in function of the distance from the inner side (occlusal side) of the element (fixed effect) with transect and the side (inner, outer or tip) as random effects (Fig. 5; Tables S1 and S2). To account for different thicknesses of the crown tissue in different parts of the element, the length of each transect was scaled to the [0, 1] interval (*Leonhard et al., 2021*).

## Analysis of growth dynamics

High resolution BSE photographs were produced using Helios NanoLab 600i field emission FIB SEM at the Department of Materials Science, Friedrich-Alexander-Universität Erlangen-Nürnberg at 15 kV. One transect has been placed along one transect in each specimen (Figs. 2 and 3). Growth layers were counted and measured using the measurement tool in ImageJ (Fiji). This was conducted on both elements on the convex side, which is assumed to be the non-occlusal side of the element (Figs. 2 and 3). We attempted to fit von Bertalanffy and logistic growth models using the package

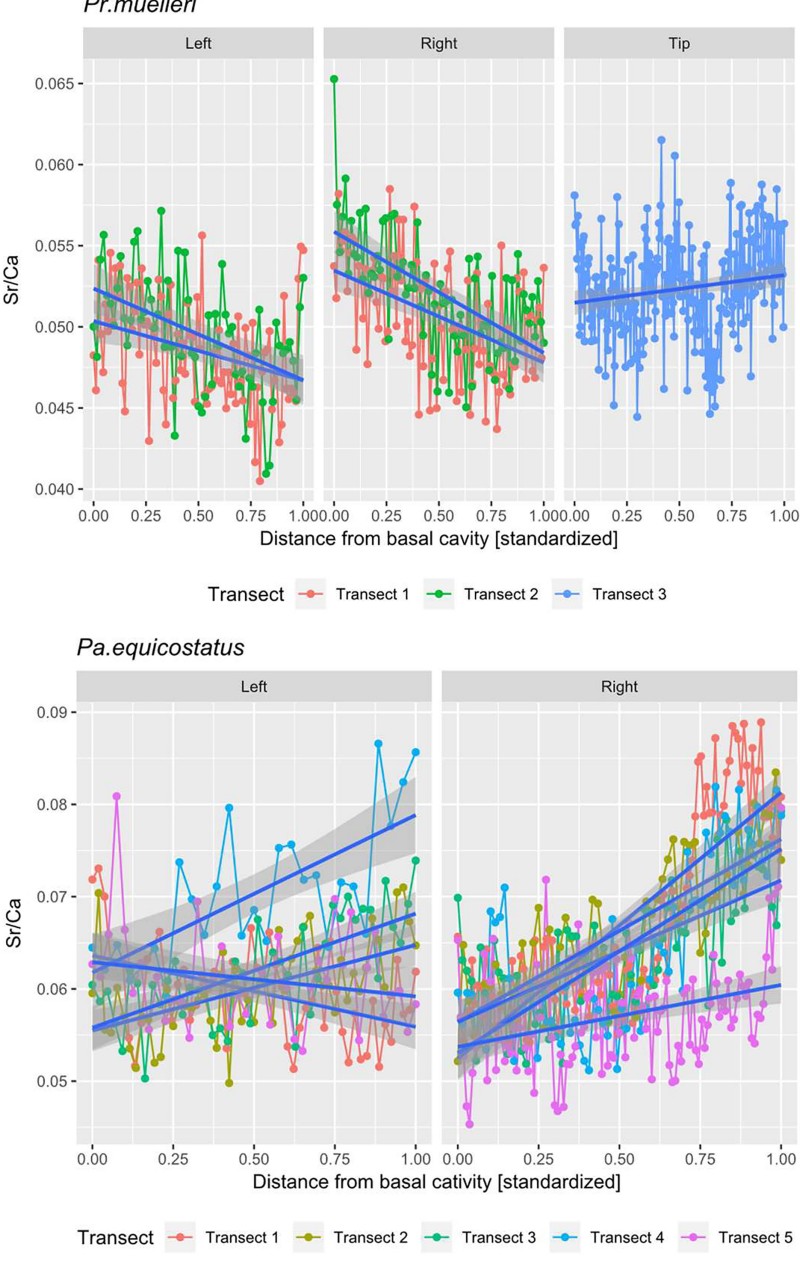

**Figure 5 Linear mixed effect model fitted to Sr/Ca ratio across the crown tissue of *Panderodus equicostatus* and *Proconodontus muelleri*.** Linear mixed effect model fitted to Sr/Ca ratios across transects 1–3 through crown tissue of *Pr. muelleri* and transects 1–5 through the crown tissue of *Pa. equicostatus* from the inner to outer side of the element. The length of each transect was scaled to the [0, 1] interval.

growthrates, but these models could not be fitted to the data (see "Discussion"). An OLS linear growth model was compared with an allometric model fitted using the nls function with a self-starter from the aomisc package (*Onofri, 2020*) or using the drc package (*Ritz et al., 2015*). Model selection was based on Akaike's Information Criterion (AIC). The results are reported in Table 1 and in Figs. 6B and 7B.

**Table 1 Growth models fitted to cumulative growth curves of both specimens.**

| Taxon | Model descriptor | Linear growth model | Allometric growth model |
|---|---|---|---|
| *Panderodus equicostatus* | Formula | $y = 0.3941x + 0.4725$ | $y = 0.6092x^{0.8749}$ |
| | Residual standard error | 0.2498 | 0.1554453 |
| | AIC | 5.640348 | −19.97418 |
| *Proconodontus muelleri* | Formula | $y = 0.5472x + 0.8114$ | $y = 0.7891x^{0.9119}$ |
| | Residual standard error | 0.5721 | 0.3597 |
| | AIC | 103.7894 | 49.95944 |

**Note:**
Models fitted to cumulative growth curves obtained from BSE images across sections through *Panderodus equicostatus* (25 degrees of freedom) and *Proconodontus muelleri* (56 degrees of freedom). All model parameter estimates were significant at alpha = 0.001.

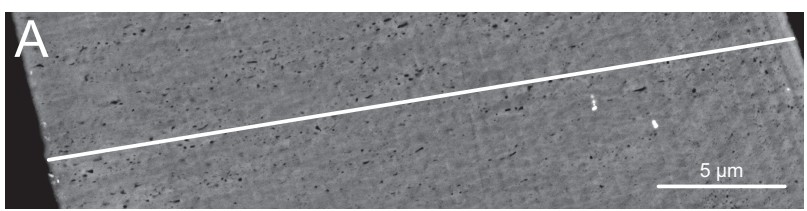

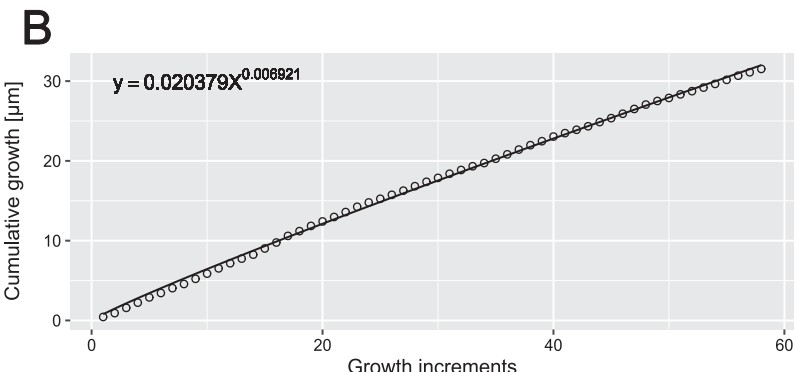

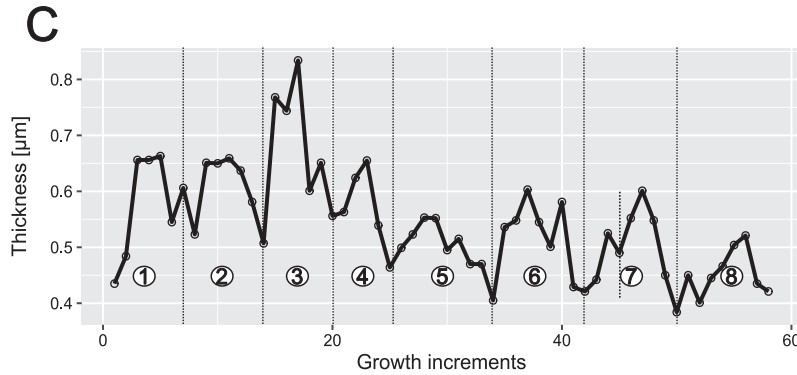

**Figure 6 Reconstruction of the growth dynamics of *Proconodontus muelleri*.** Reconstruction of growth dynamics of *Pr. muelleri* obtained from the high-resolution BSE image (A). (A) Transect through lamellar tissue on the outer side of the element along which lamellae were counted from the inner side towards the outer edge; (B) growth curve with power law function fitted; (C) thickness [µm] and number of counted growth increments; Deposition of 48 increments with mean width of 0.47 µm in six bundles of seven-nine increments each.

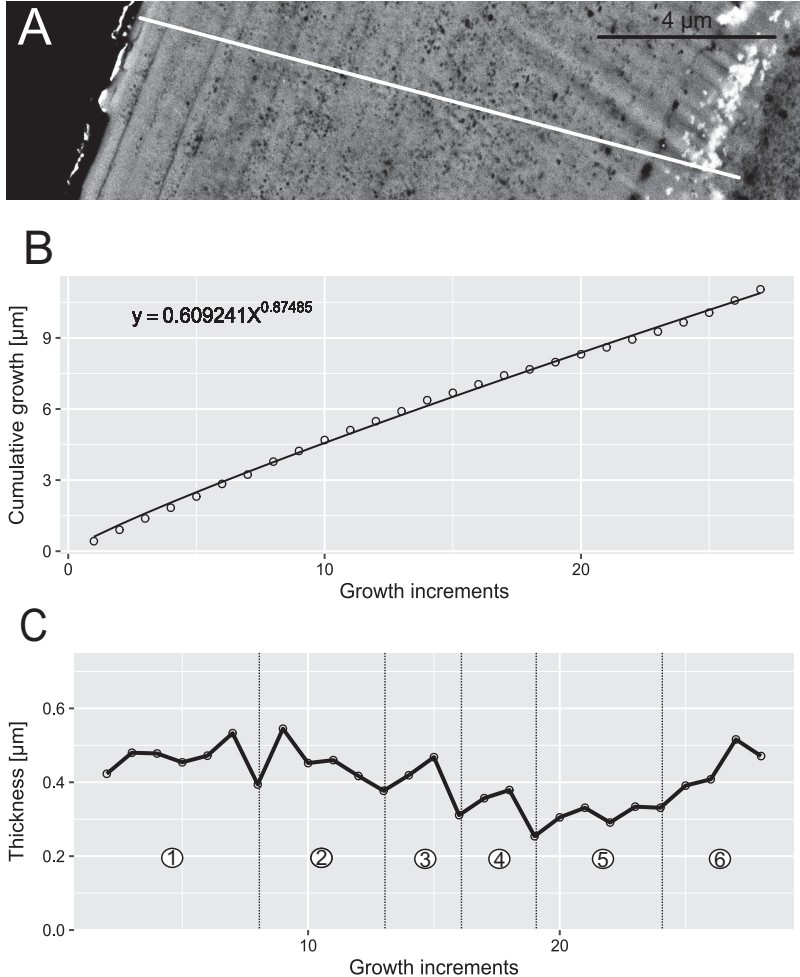

**Figure 7 Reconstruction of the growth dynamics of *Panderodus equicostatus*.** Reconstruction of growth dynamics of *Pa. equicostatus* obtained from high-resolution BSE image (A). (A) Transect through lamellar tissue on outer side of the element along which lamellae were counted from the inner side towards the outer edge (lamellae adjacent to the basal body were not clearly detectable); (B) growth curve with power law function fitted; (C) thickness [μm] and number of counted growth increments; Deposition of 22 increments with mean width of 0.37 μm in six bundles with 4–7 increments each.

## RESULTS

### Growth dynamics

In both specimens, allometric growth models described the growth curves better than linear models (Table 1; Figs. 6B and 7B). Growth layers adjacent to the basal body in the *Pa. equicostatus* specimen were not clearly detectable (Fig. 7A). Further from the inner side of the crown 27 (minimum estimate) growth layers with a mean width of 0.44 μm were counted. We measured an average of accretion of 13.10 μm of the lamellar tissue on the non-occlusal side (outer side) of the element. Growth dynamics followed an allometric model $y = 0.6092x^{0.8749}$ (Table 1). The individual grew faster in the first third of its life until growth layer eight (mean width = 0.472 μm), slowed down until growth layer 22 (mean width = 0.368 μm) and increased the speed of growth towards the edge of

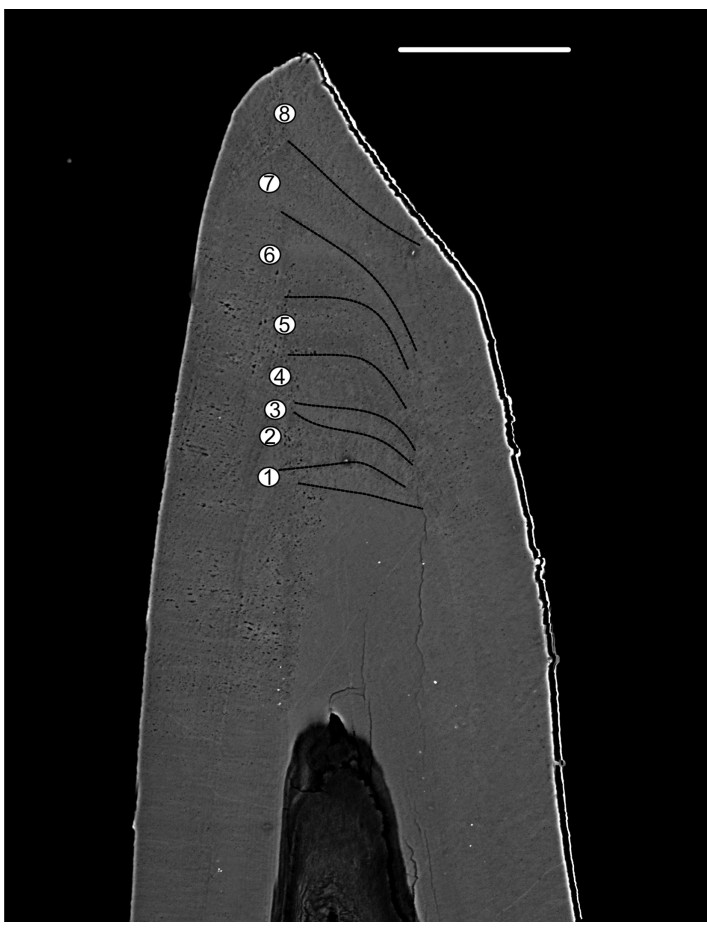

**Figure 8 BSE image of the tip of *Proconodontus muelleri*.** High-resolution BSE image of the tip area of *Pr. muelleri*. Sequence of eight truncated, irregular surfaces (scale bar 35 μm).

the element (Fig. 7B; mean width = 0.423 μm). We observed a periodicity (deposition of growth increments in bundles) with 4–7 growth increments in six bundles (Fig. 7C).

*Pr. muelleri* had 58 growth layers (Fig. 6A) with a mean width of 0.543 μm on the non-occlusal side of the element (accretion of 29.579 μm in total). The specimen followed allometric growth described as $y = 0.7891x^{0.9119}$ (Table 1). It grew faster within the first third of its life until growth layer 18 (Fig. 6B; mean width = 0.62 μm), then it continued to grow at a constant rate before it slowed down at growth layer 48 (mean width = 0.47 μm) towards the end of its construction. *Pr. muelleri*'s growth showed periodicity with 7–9 growth increments per bundle (Fig. 6C). We counted eight bundles of growth increments, which matched with the sequence of eight truncated, irregular surfaces detected within the tip of the specimen (Fig. 8).

## Surface damage

No internal wear or damage surfaces could be detected in BSE sections through either of the specimens. White matter, expected in the cusp of *Pa. equicostatus*, was not detected under light microscope (Fig. 1A) or BSE (Fig. 3). *Pr. muelleri* elements lack white

matter. In the *Pa. equicostatus* specimen, the lamellar tissue present in the tip area had porous interlamellar zones such as those described by *Müller & Nogami (1971)*.

### Sr/Ca ratio

Mean ± SD strontium content in *Pa. equicostatus* ranged from 206 ± 23 (transect 5) to 296 ± 42 (transect 1) cts and calcium content—from 3,561 ± 212 (transect 5) to 4,593 ± 166 (transect 1) cts. In *Pr. muelleri*, Sr content ranged from 321 ± 22 (transect 1) to 540 ± 30 (transect 3) cts and Ca content—from 6,346 ± 284 (transect 2) to 10,323 ± 140 (transect 3) cts. The relative concentrations per transect are summarized in Fig. 4. Systematic differences between transects have been accounted for in the design of the random intercept mixed-effects model, allowing to compare the results across the entire element.

The mixed effect model fitted to Sr/Ca ratios across five transects placed through crown tissue of *Pa. equicostatus* indicated an increase of Sr/Ca during ontogeny and high variance between individual transects. On the inner side of the element, all transects showed an increase, whereas on the outer side, two out of the five examined transects recorded a decrease, reflecting the high variability of the slopes (Table S2). We excluded transect six from the model since most of the measured counts of Sr and Ca are lying within the basal body (Fig. 3). The fixed effect of the distance from the inner edge of the lamellar crown tissue was estimated as $y = 0.011792x + 0.056889$ ($n = 647$, standard error for the intercept estimate 0.002815 and standard error for the slope estimate 0.007881). The rate of increase was different on either side of the element, with higher values on the inner side (slope coefficient 0.019) than on the outer side (0.004). The strongest increase in Sr/Ca was found along transect 4 (Table S2).

In *Pr. muelleri*, the mixed effects model fitted to Sr/Ca ratios across three transects running through the lamellar tissue (Figs. 2 and 5) indicated a decrease during ontogeny (Table S1), with the fixed effect of the distance from the inner edge of the crown estimated as $y = -0.003149x + 0.052491$ ($n = 572$, standard error for the intercept estimate 0.001396, standard error for the slope estimate 0.002627). The side of the element had a greater effect on the intercept estimate, with the highest values on the inner side (Table S1), whereas the slope was more affected by the position of the transect, with an opposite sign (increasing Sr/Ca) along transect 3 running through the tip of the element and with the strongest decrease along transect 2. Apart from the weak increase of Sr/Ca values through the tip of the element, individual estimates calculated for each level of the random effects did not differ substantially from that of the fixed effect.

## DISCUSSION

### Element growth in *Proconodontus muelleri* and *Panderodus equicostatus*

Neontological theories of growth dynamics rely on information on resource availability and the network of interactions in which the organism engages; this information is not available for most fossils. Thus, interpretations of growth dynamics must rely on theoretical models. Here we attempt to apply simple predictions of optimal resource

allocation models to interpret growth curves of *Pa. equicostatus* and *Pr. muelleri*. These models aim to predict the distribution of energy into somatic growth and reproduction, taking into account what size of the adult animal and of its offspring allows for the best resource acquisition and the lowest mortality. Here we observed indeterminate growth, *i.e.* growth throughout the life of the individual (*Lincoln, Boxshall & Clark, 1982*; *Sebens, 1982*). We can exclude that the specimens were immature because their measured element length (*Pa. equicostatus*: 725 μm; *Pr. muelleri*: 782 μm) is in the upper third of average element length of most illustrated specimens of the same species in recent literature (Supplemental Material: Fig. 1). Thus, we can be certain that their growth, which was best described by a power equation, but close to linear, was representative of their life history and not limited to the linear growth phase, which is characteristic *e.g.* for fish before they reach sexual maturity (*Sebens, 1987*). The strongest theoretical predictor of indeterminate growth is seasonality (*Kozlowski, 2006*), because seasonality leads to a periodic shift in the benefits of growth and reproduction and incentivises repeated episodes of resource accumulation to reproduce in successive seasons. Growth dynamics similar to that induced by seasonality can be caused by design constraints, *e.g.* limits on the space for egg development in the body cavity or time needed for tissue maturation (*Stamps, Mangel & Phillips, 1998*; *Ricklefs, 2003*). We suggest that conodonts were likely annual, multivoltine organisms, *i.e.* reproducing multiple times during the season. Periods of feeding and somatic growth alternating with periods where energy is allocated to reproduction are expected to lead to growth described by the logistic or von Bertalanffy's curves (*Kozlowski, 2006*). Distinguishing between these models was not possible with the proposed short conodont life spans examined here, but asymptotic indeterminate growth is seen in "complex" conodonts (*Dzik, 2008*; *Shirley et al., 2018*). The difference between the nearly-linear growth found in coniform conodonts described here and that characterized by a strong decrease in the growth rate *e.g.* in *Oz. confluens* (*Shirley et al., 2018*) or in the *Tripodellus* lineage (*Dzik, 2008*) is consistent with a stronger top-down control: slowing growth with age indicates allocating bigger fraction of energy into reproduction, which is an optimal strategy if mortality increases with size (*e.g.* when large predators are present in the ecosystem). Nearly linear growth as observed in *Pr. muelleri* and *Pa. equicostatus* is predicted to be optimal if size allows for more offspring without the risk of increased mortality, *i.e.* when growing bigger does not mean becoming an easier prey. Based on these simple predictions of optimal resource allocation models, it may be possible to exploit conodont sclerochronology to identify the evolution of life histories as early Palaeozoic trophic networks became more complex.

We recognize the limitations of a sample size of two elements from two taxa, and the discussion below is in the context of the degree to which these data can be extrapolated to other elements and taxa and the confidence we can place in the resulting conclusions.

## Damage and wear

Complex conodont elements exhibit repeated episodes of wear and damage within their crown tissues, indicating dental function (*Purnell, 1995*; *Donoghue & Purnell, 1999a*; *Purnell & Jones, 2012*; *Shirley et al., 2018*). In the few taxa known from clusters, the

distribution of wear and damage appears to match the occlusal contact between pairs of elements within the apparatus (*Donoghue & Purnell, 1999a*, *1999b*). A limitation of studies available so far is that patterns of damage are either described on element surfaces or in histological sections, but these two aspects—surface and internal structure—have not been compared in the same element. Consequently, of the various types of wear and damage identified by *Purnell & Jones (2012)*, it is mostly breakage that has also been identified in histological sections, including in coniform conodonts (*Hass, 1941*; *Barnes, Sass & Monroe, 1970*; *Müller & Nogami, 1971*; *Barnes, Sass & Poplawski, 1973*; *Nazarova & Kononova, 2020*). On the element surface, breakage can only be detected when it has been so extensive that the original shape of the element could not be fully restored. In such cases the tips of cusps appear smaller in diameter where regeneration has taken place. Smaller-scale wear and damage, such as polishing and rounding described from surfaces of "complex" conodonts by *Purnell & Jones (2012)*, are not likely to leave any trace if they had already been covered by the next episode of tissue deposition. We are not aware of any reports of such damage in coniform conodonts.

## Implications for other types of elements in the apparatus

*Murdock, Sansom & Donoghue (2013)* documented morphological and functional specialization of elements in the *Panderodus* apparatus and characterized the anterior (costate) element suite as represented by larger elements that are most resistant to bending and torsion. However, the re-examination of the feeding apparatus of *Panderodus* by *Murdock & Smith (2021a)* positioned the truncatiform element not within the costate suite, but in the compressed, posterior suite. Its new position does not alter the examined resistance to bending and torsion and has no effect on its functional specialization.

Truncatiform elements examined here are more resistant to bending in one direction, which is interpreted to be characteristic of element types which functioned as blades and were used for cutting prey items. Truncatiform elements of *Panderodus* have been also placed by *Murdock, Sansom & Donoghue (2013)* in the functional unit of the apparatus characterised by the lowest resistance to torsion, which would correspond to elements not involved in prey restraint (*Murdock & Smith, 2021b*: https://datadryad.org/stash/dataset/doi:10.5061/dryad.0p2ngf20z). Elements with a cutting function, *i.e.* arcuatiform, truncatiform and falciform, would be the most likely candidates to show surface wear and damage; the fact that we could not identify it in a truncatiform element suggests that this lack is representative for the entire apparatus of *Pa. equicostatus*.

The apparatus of *Proconodontus* has lower morphological and, most likely, lower functional differentiation than *Panderodus* (*Sansom, Armstrong & Smith, 1994*). The apparatus of *Pr. muelleri* has been reconstructed as a multielement apparatus (*Szaniawski & Bengston, 1998*) based on discrete elements only. We could not identify undebatable wear or damage surfaces in the aequaliform element of this species. If the functional interpretation developed for *Panderodus* by *Murdock, Sansom & Donoghue (2013)* is applied to *Pr. muelleri*, the symmetry of aequaliform elements likely results in an equally distributed resistance to bending in all directions. This property was interpreted in *Panderodus* as an adaptation for prey capture and restraint (*Murdock, Sansom &*

*Donoghue, 2013*). It is possible, therefore, that wear and damage could be present only in elements interpreted to perform cutting function, *i.e.* arcuatiform, truncatiform and falciform. In *Panderodus*, the aequaliform element was placed by *Sansom, Armstrong & Smith (1994)* in the posterior-most functional unit with the lowest average resistance to torsion, which was therefore interpreted as not involved in restraining the prey item (*Murdock, Sansom & Donoghue, 2013*). However, regarding the most recent reconstruction by *Murdock & Smith (2021a)* for the *Panderodus* apparatus, the aequaliform element was exposed on the midline between the anterior and posterior suite.

But symmetrical elements, *i.e.* truncatiform and aequaliform, have relatively high resistance to torsion. Furthermore, morphological specialization of element types in *Proconodontus* apparatuses appears to be less pronounced, with many transitional forms reported within single populations (*Szaniawski & Bengston, 1998*). Consequently, in a less specialized apparatus likely all elements engaged with the prey items and their functions overlapped to a larger extent than they did in *Panderodus*.

## Implications for other conodont taxa

*Sansom, Armstrong & Smith (1994)* proposed a division of the *Panderodus* apparatus into three functional units, applicable to all species. Subsequent analysis of individual morphologies proposed that this division did not capture the degree of specialization of individual elements (*Murdock et al., 2013*). Shape features which formed the basis of this functional analysis are largely preserved and recognizable across the genus and are, in fact, used to identify element types (*e.g. Sansom, Armstrong & Smith, 1994*; *Jeppsson, 1997*). Finite Element Analysis comparing *Proconodontus*, *i.e.* conodonts with crown tissues, and the paraconodont *Furnishina*, devoid of crown tissues, indicate that histological differentiation has an even larger impact on mechanical properties than the shape alone (*Murdock, Rayfield & Donoghue, 2014*). Panderodus species differ systematically in the proportion of crown tissues and the depth of their basal cavity (which is *in vivo* filled with dentine-like basal tissue) and these systematic differences are the basis for species diagnoses. In particular, some species such as *Panderodus panderi*, have a much higher proportion of white matter than *Pa. equicostatus* examined here. As the mechanical analysis by *Murdock, Sansom & Donoghue (2013)* relied exclusively on element outlines and not on the histological composition, it is likely that the functional differentiation within the apparatus of any given *Panderodus* species would remain the same, regardless of species-specific histological differences.

Elements of *Proconodontus* differ in the inner structure (basal cavity to crown ratio) and their morphology between species. The genus is one of the very early and primitive euconodonts with striking morphological similarities to their ancestors (paraconodonts) but with a crucial apomorphy, the crown tissue (*Müller & Hinz-Schallreuter, 1998*; *Murdock, Rayfield & Donoghue, 2014*). The expansion of the crown and the simultaneous reduction of the basal body may be one of the leading factors towards high morphological and functional diversity in euconodonts, leading to their great diversity in feeding ecology (*Murdock, Rayfield & Donoghue, 2014*). Elements of *Pr. muelleri* exhibit relatively

deep basal cavities (Fig. 1), whereas other species feature proportionally thicker crowns, presumably to distribute stress more evenly while functioning (*Jones et al., 2012b*).

## Growth periodicity

Growth layers were grouped into bundles of 4–7 and 7–9 in, respectively, *Pa. equicostatus* and *Pr. muelleri*. These bundles are shorter than the "major increments" with averages of 16–17 increments, observed in coniform conodonts *Protopanderodus varicostatus* and in *Drepanodus robustus* by *Armstrong & Smith (2001)*.

Correlation between bundles of growth increments and periods of dental function visible as damage on the occlusal surface observed in *Oz. confluens* were interpreted by *Shirley et al. (2018)* as support for the model of conodont growth in which periods of element use corresponded to growth arrest and alternated with element repair (*Bengtson, 1976*; *Zhan, Aldridge & Donoghue, 1997*). On the other hand, the lack of damage and repair found in this study suggests that, at least in these elements, periodicity was present even in the lack of repair periods. These bear a resemblance to Retzius Periodicity, which is driven by an unknown internal biorhythm and is not associated with any functional periodicity, marking bundles of daily cross-striations in mammals (*Boyde et al., 1989*; *Antoine, Hillson & Dean, 2009*; *McFarlane et al., 2021*). In such case, the repair cycle following the circaseptan growth periodicity might have been an exaptation.

Assuming no periods of growth arrest during each cycle, *i.e.* if the number of growth layers is taken at face value, the life span of early coniforms, *Pr. muelleri*, *Protopanderodus varicostatus* and *Drepanodus robustus*, appears to be longer than that of the derived genus *Panderodus* examined here, as well as that of "complex" conodonts. Those studies which examined multiple specimens per sample reported small intraspecific variation, suggesting that values obtained here are representative for the respective species.

## Trophic shifts during ontogeny

Sr/Ca ratio analysis of skeletal tissues has been applied previously to investigate palaeodiets and relative positions of animals within the trophic network (*Balter et al., 2002*; *Peek & Clementz, 2012*). However, environmental conditions (*e.g.* water chemistry, temperature or salinity) can affect these ratios (*e.g. de Villiers, 1999*; *Zimmerman, 2005*; *Martin & Thorrold, 2005*): changes in the Sr content and/or Sr/Ca ratio in skeletal tissues can potentially reflect migration of the organism over distances or across the temperature gradient within the water column (*de Villiers, 1999*; *Shirley et al., 2018*).

Based on the sedimentological record (see geological setting in *Jarochowska et al., 2016*), *Pa. equicostatus* lived on a shallow carbonate platform without a substantial temperature gradient. *Pr. muelleri* has been interpreted as having had a pelagic mode of life (*Miller, 1984*), therefore it probably stayed within the surface waters above the thermocline. We are, therefore, confident that the Sr/Ca ratios observed here can be attributed to trophic level rather than fluctuating environmental conditions.

Contrary to expectations based on "complex" conodonts, chemical proxies for the trophic position did not indicate changes in this position during the ontogeny of *Pr. muelleri* and *Pa. equicostatus*. The Sr/Ca ratio, proposed as a proxy for the trophic

position, changed only minimally throughout the ontogeny in both species and in opposite directions: it decreased, as predicted based on comparisons with *Ozarkodina confluens* (*Shirley et al., 2018*), in *Pr. muelleri*, but increased by ca 1% per 1 µm in *Pa. equicostatus*. These results indicate that neither species changed their trophic niche substantially during their life. The study by *Shirley et al., 2018* was the first one investigating Sr content during ontogeny of the "complex" conodont *Ozarkodina confluens*. The decrease in Sr contents in crown tissue (*Shirley et al., 2018*) was attributed to "biopurification" in the trophic network and was in this species coincident with the appearance of histological record of damage along occlusal surfaces in the adult animal. During its early life, the animal fed at a lower trophic level than the adult, which was interpreted to have adopted a predatory or scavenger lifestyle. Here we refined this proxy by measuring the Sr/Ca ratio and not the Sr contents. This should not render comparison impossible, because Sr is the most common ion to replace Ca in the francolite lattice.

Our results indicate that either species did not change their trophic niche substantially during their life, coinciding with almost linear growth. Lack of trophic differentiation in these species is consistent with lack of tissue damage which could record dental function. Concurrent chemical and histological observations do not yield any evidence for direct occlusal and predatory habit of *Pr. muelleri* and *Pa. equicostatus* which has been demonstrated for "complex" conodonts (*Purnell, 1995*; *Jones, 2009*; *Jones et al., 2012a*; *Martínez-Pérez et al., 2016*; *Shirley et al., 2018*).

## CONCLUSIONS

We used two coniform conodont elements, the phylogenetically primitive late Cambrian *Proconodontus muelleri* and the more derived Silurian *Panderodus equicostatus*, to test the hypothesis whether their adult forms fed as predators or scavengers. Unlike in "complex" conodonts, no damage of the crown tissue, which is indicative of dental function, could be detected in histological sections. An independent chemical proxy, the Sr/Ca ratio, which was expected to decrease with the trophic level, did not indicate shifts in the trophic position in the two examined specimens Growth increments formed bundles of 4–7 in the crown tissue of *Pr. muelleri* and 7–9 in *Pa. equicostatus*, respectively, which we interpret as driven by an internal clock and analogous to Retzius Periodicity in vertebrate teeth. This finding contradicts our previous interpretation that periodicity was an adaptation to tissue repair following damage (*Shirley et al., 2018*) and indicates that the circaseptan rhythm was present in conodonts even in the absence of tissue damage during feeding periods. Internal periodicity is consistent with indeterminate growth in conodonts when interpreted in the context of optimal resource allocation models (*Kozlowski, 2006*). Repeated periods of growth would shift resource allocation away from reproduction. Although conodont growth dynamics have not been investigated systematically, growth curves in *Pa. equicostatus* and *Pr. muelleri* do not have strong asymptotes as observed in "complex" conodonts (*Dzik, 2008*; *Shirley et al., 2018*). Such growth dynamics is predicted to be optimal where there is no size-dependent increase in mortality, such as in the absence of larger predators which would preferentially target large individuals. The limitations of our study are that we examined only a single element

and only one type of element in each apparatus: truncatiform in *Panderodus equicostatus* and aequaliform in *Proconodontus muelleri*. Given functional differentiation of these apparatuses (*Sansom, Armstrong & Smith, 1994*; *Murdock, Sansom & Donoghue, 2013*; *Murdock & Smith, 2021a*), it is not certain whether the lack of tissue damage is representative of the entire apparatus. Chemical and sclerochronological records, however, are expected to be consistent for the entire individual. Our study suggests that trophic ecology of coniform conodonts in early Palaeozoic ecosystems differed from that of predators or scavengers documented for "complex" conodonts. Our results suggest also that conodonts underwent an evolution of their life histories towards a top-down control consistent with the appearance of large predators by the Silurian Period (*Klug et al., 2017*).

## ACKNOWLEDGEMENTS

We thank Birgit Leipner-Mata for help in preparation of sections and Christian Schulbert for help with SEM. We thank J.D. Loch and J.F. Taylor for obtaining the Windfall samples. We thank the Federica Marone (Swiss Light Source, Paul Scherrer Institut, Villigen, Switzerland) for access to the beamline and for invaluable assistance in collecting SRXTM data. Any use of trade, firm, or product names is for descriptive purposes only and does not imply endorsement by the U.S. Government. The manuscript benefited from constructive reviews by Marc Leu, Annalisa Ferretti and Yanlong Chen.

### Funding

This work has been supported by the Deutsche Forschungsgemeinschaft (Grant No. Ja 2718/1-3). The funders had no role in study design, data collection and analysis, decision to publish, or preparation of the manuscript.

### Grant Disclosures

The following grant information was disclosed by the authors:
Deutsche Forschungsgemeinschaft: Ja 2718/1-3.

### Competing Interests

The authors declare that they have no competing interests.

### Author Contributions

- Isabella Leonhard conceived and designed the experiments, performed the experiments, analyzed the data, prepared figures and/or tables, authored or reviewed drafts of the paper, and approved the final draft.
- Bryan Shirley conceived and designed the experiments, performed the experiments, analyzed the data, prepared figures and/or tables, authored or reviewed drafts of the paper, and approved the final draft.
- Duncan J. E. Murdock conceived and designed the experiments, performed the experiments, authored or reviewed drafts of the paper, and approved the final draft.

- John Repetski analyzed the data, authored or reviewed drafts of the paper, and approved the final draft.
- Emilia Jarochowska conceived and designed the experiments, performed the experiments, analyzed the data, prepared figures and/or tables, authored or reviewed drafts of the paper, and approved the final draft.

## Data Availability

The data is available at OSF: Leonhard, Isabella, Emilia Jarochowska, Bryan Shirley, Duncan Murdock, and John Repetski. 2021. "Growth and Feeding Ecology of Coniform Conodonts." OSF. November 3. osf.io/9npz2.

The Morphobank project is available at: https://morphobank.org/index.php/Projects/ProjectOverview/project_id/3589

The Panderodus equicostatus specimen is stored as an SEM mount in the collections of GeoZentrum Nordbayern, Friedrich-Alexander-Universität Erlangen-Nürnberg, at Loewenichstr. 28, 91054 in Erlangen, Germany:

EJ-12-V-19.25-001.

## Supplemental Information

Supplemental information for this article can be found online at http://dx.doi.org/10.7717/peerj.12505#supplemental-information.

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
