# Peer review of "Growth and feeding ecology of coniform conodonts"

_PeerJ, doi:10.7717/peerj.12505_

## Round 0.1 · original submission · Minor Revisions

I have now received reports from three reviewers and, after careful consideration, I have decided to invite a minor revision of the manuscript. Collectively, the Reviewers think this paper is interesting and significant and should be published in PeerJ. As you will see from the reports copied below, the reviewers raise important concerns. If you feel that you are able to comprehensively address the reviewers’ concerns, please provide a point-by-point response to these comments along with your revision. If you are unable to address specific reviewer requests or find any points invalid, please explain why in the point-by-point response.

·

Basic reporting

For: PeerJ
There are no significant problems with the science of this submission as the article is self-contained with relevant results to the hypotheses. It is very well-written in a professional English throughout the manuscript and well-organized. The literature references are sufficient and up-to-date. The minor problems are the following:
1. Generally, the citation format could be more consistent. Sometimes it is written “and” and sometimes “&”. E.g. Murdock and Smith, 2021 (line 83) but Szaniawski & Bengtson 1998 (line 100)

2. line 84: publication year of reference is incorrect.

3. There is a confusion in the manuscript referring to the citation of Murdock & Smith 2021. In the reference list are two references from Murdock & and Smith 2021, (Murdock & Smith 2021a) and (Murdock & Smith 2021b). This should be marked the same way throughout the manuscript.

4. line 477: “exaptation” spelling error

5. lines 488-492 and lines 500-504. There is a repetition of almost the same sentences.

“The Sr/Ca ratio, proposed as a proxy for the trophic position, changed only minimally throughout the ontogeny in both species and in opposite directions: it decreased, as predicted based on comparisons with Ozarkodina confluens (Shirley et al., 2018), in Pr. muelleri, 491 but increased by ca 1% per 1 µm in Pa. equicostatus. These results indicate that neither species changed their trophic niche substantially during their life”
and
“The Sr/Ca ratio changed only minimally throughout the ontogeny in both species and in opposite directions: it decreased, as predicted based on comparisons with Ozarkodina confluens (Shirley et al., 2018), in Pr. muelleri, but increased in 503 Pa. equicostatus. These results indicate that either species did not change their trophic niche substantially during their life”
I suggest some rephrasing.

Experimental design

The research question is well defined, relevant and meaningful. This research fills an important knowledge gap. No previous studies on conodonts examined the trophic levels and the lifestyles of conodonts by using Sr/Ca ratios. Furthermore there are only very limited studies on growth rates in primitive coniform conodonts. The technical standards used in this study are state-of-the-art with the newest techniques available, e.g. backscatter electron (BSE) imaging, energy-dispersive X-Ray spectroscopy (EDX) and synchrotron radiation X-ray tomographic microscopy (SRXTM).

1. Lines237-238: The authors attempted to fit von Bertalanffy and logistic growth models using the package growth rates, but these models could not be fitted to the data. There should be an additional sentence or add on to find the explanation in the discussion (e.g. (see discussion)).

Validity of the findings

The results of this study are significant because the using Sr/Ca ratios for paleodiets in aquatic systems is a novelty and makes an important contribution to the field. The conclusions are well stated and linked to the original research questions. The minor problems are the following:
1. line 324: The claim that most conodonts lived less than a year without any citation is mainly based on assumptions that lamellae in conodont crown tissues represent regular daily episodes of depositions. To my knowledge there is not enough evidence supporting this claim. With no similar living present representatives of conodonts, it is highly speculative to impose circadian rhythms from mammals and reptiles on conodonts. Maybe one or two sentences on which basis the implementation was made would help the reader to better understand.

2. lines 327-328: To my knowledge there is no data or evidence available concerning the reproduction in conodonts. Thus the claim that condonts reproduced multiple times during a season is highly speculative. Maybe more information to support your sentence is needed.

Additional comments

The main issues of this study are the small amount of investigated specimens (2). Whether the elements are representative for the whole animal or even the entire population can be neglected for the Sr/Ca ratio studies and the investigations on the growth rates. But the damage surface of the element can be highly individual and it has to be questions if the lack of crown tissue damage in two elements are enough evidence to suggest the trophic ecology of coniform conodonts in early
Palaeozoic ecosystems in general. However, the authors support their findings to a certain degree with independent Sr/Ca studies.
The interpretation of the Sr/Ca ratio data needs more detail. De Villiers (1994; 1999) documented and increase of the Sr/Ca values with water depth, there is a minute depth gradient (in modern oceans). Furthermore, the Sr/Ca values differ latitudinal and longitudinal (in modern oceans). Your interpretation is mainly based as if the conodonts were living always on the same spot. I am missing in the discussion the possibility of conodonts changing the water depth during their live or that they migrated. How would this influence the Sr/Ca values? Different Sr/Ca values in different parts of the paleo-waters should be more elaborated in the Discussion.

In summary, this study should be published but only after the minor revisions mentioned before.

·

Basic reporting

I think this is a really interesting contribution to our understanding of the feeding ecology of conodonts, specifically addressed to the conodont group bearing coniform elements that is usually hardly investigated. A few suggestions are reported below.

Experimental design

I appreciated the clarity of the exposure.
The only two minor comments that you might consider having the authors address would be the need to specify for non-conodont people what euconodonts are (line 113) and perhaps discuss/reply to those who have recently questioned geochemical stability of bioapatite, i.e. Shohel et al. (2020, R. Soc. Open Sci. just using coniform elements of Dapsilodus obliquicostatus) or Ferretti et al. (2021, Palaeo3).

Validity of the findings

My only concern is that all data derive from the study of two only elements, one for each genus. Maybe the authors should discuss how confident their result may be.

Additional comments

A few minor comments regard the need of:
- line 84: 2021 instead of 202;
- line 86: replace assigned with a synonym (assigned used also in previous line);
- add commas before year of publication (e.g., lines 100, 120, 123, 198., ...);
- remove commas before et al. (e.g., lines 127, 184, 201, ...);
- remove and at the end of line 139;
- line 165: late and not Upper Cambrian;
- be consistent in using numbers in arabic (e.g., line 255: 4-7 growth increments) or written in full (four-seven in explanation of figure 7).

·

Basic reporting

The manuscript was well written and well organized.

Experimental design

The research goal has been well achieved.

Validity of the findings

All these data, as well as the method, are tremendous important for our understanding of the evolutionary of conodonts. Thus I strongly support its publication in the journal.

Additional comments

Dear editor and author,
This interesting article addresses the growth and feeding ecology of conodonts using histological and geochemical methods. In general, the manuscript was well written and well organized. The research goal has been well achieved. All these data, as well as the method, are tremendous important for our understanding of the evolutionary of conodonts. Thus I strongly support its publication in the journal.
However, there are also some problems, especially see my comments on conodont life spans, which should be carefully considered before its publication. My comments and suggestions are as follow.
With best regards,
Yanlong Chen
* * *
Northwest University, China; email: yanlong.chen@nwu.edu.cn
* * *
Line 57: may add reference: Joachimski et al., 2002, 2009
Line 129: may add reference: Chen et al., 2016
Line 144: the “formula (Ca5Na0.14(PO4)0.016F0.73(H2O)0.85)” is problematic. It should be changed as “Ca5Na0.14(CO3)0.16(PO4)3.01(H2O)0.85F0.73”, and a reference may added: Joachimski, M. M., Breisig, S., Buggisch, W., Talent, J. A., Mawson, R., & Gereke, M., Morrow J.R., Day J., Weddige K., 2009. Devonian climate and reef evolution: insights from oxygen isotopes in apatite. Earth & Planetary Science Letters, 284, 599–609.
Line 519: In the “Conclusions” part. The life span of 27 and 58 days is not well discussed and supported in the “Discussion” part. To my knowledge, it is still debate whether each lamina is the result of a daily, weekly, seasonal or annual secretion (e.g. Muller & Nogami 1971; Zhang et al. 1997; Dzik 2008). Though it is very similar to enamel of mammals, but there is no evidence to support that conodont lamina is daily, as there is large difference between conodonts and mammals. Even if it is day, I am wondering if it is possible that the number of lamellae stopped increase in a potential stage when the animal is till alive? As the numbers of cross-striations of human teeth stopped increase after its eruption from gum, so the number of cross-striations does equal to the life span of the man.

---

## Round 0.2 · Minor Revisions

Thank you again for submitting your manuscript to PeerJ. Only two small issues need to be corrected. In the acknowledgements section, Chen Yanlong should be replaced by Yanlong Chen because Chen is his surname. Mahoney, P. (2020) actually published in the first issue of 2021.

---

## Round 0.3 · accepted · Accept

Thank you again for submitting your manuscript to PeerJ. The revised manuscript has met the publication criteria.